# Mental Toughness Development via Military-Style Training in the NCAA: A Three-Phase, Mixed-Method Study of the Perspectives of Strength and Conditioning Coaches

**DOI:** 10.3390/sports10060092

**Published:** 2022-06-09

**Authors:** Andreas Stamatis, Grant B. Morgan, Patrick Nyamaruze, Panagiotis Koutakis

**Affiliations:** 1Exercise and Nutrition Sciences, SUNY Plattsburgh, Plattsburgh, NY 12901, USA; 2Educational Psychology, Baylor University, Waco, TX 76798, USA; grant_morgan@baylor.edu; 3Psychology, University of KwaZulu-Natal, Durban 4041, KwaZulu-Natal, South Africa; nyamruzepatrick@yahoo.com; 4Biology, Baylor University, Waco, TX 76798, USA; panagiotis_koutakis@baylor.edu

**Keywords:** sport psychology, CSCCa, sport culture, NSCA, mentally tough, student-athlete, strength coach, military training, well-being, cultural best practices

## Abstract

Sport cultures transmit values for anticipated conduct. Recent events have resulted in injuries/deaths of National Collegiate Athletic Association (NCAA) student-athletes, usually during off-season football training. Through media reports, strength and conditioning coaches (SCC) have been allegedly involved by incorporating military-style training (MST). Mental toughness (MT) has been associated with hypermasculine subcultures in sports. For the first time, perceptions of collegiate SCCs were chosen to contribute to the development of cultural best practices in sports, via a multiphase mixed-method design (Phase 1, *n* = 465; Phase 2, *n* = 72; Phase 3, *n* = 99). Quantitative and qualitative data were collected aiming to confirm and explore the use of MST in the NCAA, its connection to SCCs, its association with MT development, and the role of the media. MST is uncommon in the NCAA. MST takes place mostly during the off-season in the form of physical, in-scope protocols while football is the most common sport. MST promotes MT. The recent media backlash is considered unfounded. Cultures promoted by SCCs do not indicate conformity of student-athletes to unethical/unhealthy expectations. Future sport psychology research and practice should continue to prioritize culture, cultural identities, and physical and mental well-being.

## 1. Introduction

Despite calls [1] that National Collegiate Athletic Association (NCAA) strength and conditioning coaches (SCCs) should be extra cautious during the 2021 collegiate football off-season in terms of *nontraumatic* (injury that is not attributed to impact/external force; examples include exertional rhabdomyolysis and the sickle cell trait) *injuries* due to physical/psychological side effects stemming from the unplanned/prolonged inactivity period (suspension of NCAA seasons), another catastrophic event took place: A 19-year old freshman football player for the Virginia Union University collapsed during off-season conditioning and later died at Virginia Commonwealth University Medical Center. This was not the first time in recent history that tragic events have resulted in serious, nontraumatic injuries/deaths of student-athletes playing for academic institutions that are members of the NCAA (e.g., the University of Maryland, the University of Houston). The majority of these kinds of incidents take place during off-season football conditioning programs [2,3,4,5,6,7,8]. Although the cause could be multi-factorial and unique to each case (e.g., primordial factors, appropriate medical coverage, acclimatization), SCCs have been allegedly implicated [9,10,11].

Professional organizations that represent SCCs and/or are directly related to that vocation have taken action. The National Strength and Conditioning Association (NSCA) and the Collegiate Strength and Conditioning Coaches association (CSCCa) created the “Joint Consensus Guidelines for Transition Periods: Safe Return to Training Following Inactivity” and both have endorsed the “The Inter-Association Task Force for Preventing Sudden Death in Collegiate Conditioning Sessions: Best Practices Recommendations”. The National Athletic Trainers’ Association (NATA) developed the “Consensus Statement: Sickle Cell Trait and the Athlete”. Additionally, the NCAA provides numerous relevant resources through their Sport Science Institute, such as “Preventing Catastrophic Injury and Death in Collegiate Athletes”. Nevertheless, the media have repeatedly turned against the purportedly hypermasculine subculture SCCs commonly initiate and/or maintain, especially in football programs, and have called for more regulation of the profession [12].

According to media reports, those SCC programs usually foster *military-style training* (MST) via visiting military facilities [13,14], organizing workshops on campus with ex-military [15,16,17], and/or incorporating MST in their own training protocols [18]. In those sorts of subcultures in sporting contexts, extreme behaviors, such as undervaluing rest [19] and overvaluing playing injured [20,21,22], are frequently rationalized and normalized. By conforming to these types of standards, MST is supposedly efficient in creating the toughest, not only from a physical but also from a psychological perspective, athletes/teams [23]. However, adopting that type of expectation without critical thinking has been identified as potentially unhealthy and unethical [24,25].

The extent of the occurrence and the magnitude of the effects of those unhealthy/unethical cultural sporting contexts becomes more evident when considering that there was a call for a global strategy to safeguard children against abusive cultures in sport [26] and a need for the development of an *overconformity* scale [27]. Although there has been evidence for the significance of exploring the sport culture further [28,29,30], there is still a need for more investigation [31] and in a more holistic way (e.g., the well-being of athletes) [32] in order to develop *cultural best practices* [22]. Among other sub-disciplines of psychology, sport psychology has been lately focusing on these socio-cultural aspects [22].

In the past two decades, *mental toughness* (“…a state-like psychological resource that is purposeful, flexible, and efficient in nature for the enactment and maintenance of goal-directed pursuits”([33] (p. 20).) MT has become a very popular sport psychology concept [34]. MT has been identified as an imperative component of athletic success [35], but also as a factor in supporting the sportsperson as a whole (e.g., transferable to social life) [36]. However, MT has also been linked to negative outcomes, especially through macrosystems (i.e., cultures) that promote subcultural demands, which may raise safety/ethical concerns, as described above.

For instance, the MT culture has been linked to machismo ideals [37], overtraining [38], emotional abuse (e.g., football) [39], ignoring pain [38,40] and vulnerability [41], cutting extreme weight (e.g., judo) [42], not seeking mental health support and being considered “weak” ([43] (p. 310)), threatening the welfare of athletes [44] and rehabilitation adherence [45], and encouraging athletes to adopt a no-excuses mentality/ethos (e.g., ultramarathoners) [46] and to sacrifice their individuality [38]. Nevertheless, MT has been found to be widespread in the US collegiate/NCAA strength and conditioning world [47,48].

The main professional organizations in that world are the NSCA and the CSCCa. The CSCCa is the only organization that claims to be exclusively dedicated to the collegiate SCC. According to them, “…the overall health and well-being of the student athlete” via evidence-based programs is the SCC’s top priority [49]. The CSCCa offers two levels of certification: (1) the Strength & Conditioning Coach Certified (SCCC) and (2) the Master Strength & Conditioning Coach (MSCC). (An MSCC is an SCCC who has been a full-time collegiate and/or professional strength SCC for a minimum of 12 years.) The latter is considered one of the highest vocational honors as it represents proficiency and durability in the field. SCCs [47] and especially MSCCs [48] have seldomly been purposely selected as participants in the sport psychology literature.

### Purpose of the Study

Based on the information presented above, the overarching purpose of this study was to contribute to cultural best practices in sports. The specific objectives included confirmation and exploration of the use of MST in the NCAA (e.g., sport, sex, period of year, specific activities), its connection to SCCs, its association with MT development, and the role of the media. The study was completed in three sequential phases following a multi-phase, mixed-method design.

In more detail, in the first phase (i.e., Study 1), there were eight guiding research questions (RQs). Based on the initial results, a second phase (i.e., Study 2) was conducted. Two emerging RQs were added. Based on the results for the first two phases, a third phase (i.e., Study 3) was deemed necessary. The number of RQs was increased to 15. All RQs are listed below per phase.

Phase 1:In terms of (a) visiting military facilities, (b) inviting on campus ex-military for workshops, and (c) SCCs incorporating any type of MST themselves, is MST popular in the NCAA?Is MST the SCCs’ initiative?Which sport(s) is/are going through MST?Is MST more popular in men’s compared to women’s sports?What time of the year is MST employed?Is MT a construct aimed to be developed through MST?In terms of MT, do SCCs think MST works?What do SCCs prescribe when incorporating MST in their training protocols?

Phase 2:9.Do the Study 1 themes match MSCCs’ understanding of how SCCs tend to use MST with their athletes?If not, what theme is missing?10.With the information collected from Study 1 in mind:What do MSCCs think most contributed to the media backlash the profession suffered?Do MSCCs think that the media backlash the profession suffered was justified?What do MSCCs think CSCCa members can do to prevent similar media backlash in the future?

Phase 3:11.Whose initiative is it to visit military facilities and/or invite ex-military on campus for workshops?12.Why is football the sport that SCCs incorporate military-style training the most?13.Without using a psychological questionnaire, how do SCCs know that their military-style training increases their athletes’ levels of mental toughness?14.Why do SCCs use military-style training more often in women’s than in men’s volleyball teams?15.Why don’t SCCs (also) use psychological interventions to increase their athletes’ mental toughness?

## 2. Materials, Methods, and Results

A three-phase, mixed-method design was chosen to examine the overarching purpose of this research through an iteration of connected studies that were sequentially aligned (see Figure 1). Study 2 confirmed and built on what was learned previously in Study 1, while Study 3 complemented the previous two studies to address the central purpose of contributing to cultural best practices’ development in the NCAA using SCCs’ perceptions. This overarching methodological framework was specifically selected due to its practical focus [50] and since the purpose could not be fulfilled with a single study as new questions emerged during the first and second phases. This strategy allowed for the implementation of two basic mixed-method designs (Study 1: explanatory sequential design; Study 2: convergent parallel design) and a purely qualitative design (Study 3).

The design of Study 1 occurred in two distinct, interactive steps. It started with the collection and analysis of quantitative data, which had the priority for addressing the RQs. This first strand was followed by the subsequent collection and analysis of qualitative data. The qualitative strand was designed to follow the results of the first one. SCCCs and MSCCs, who incorporate MST in their own training, were given the chance to describe it via an open-ended question.

The design of Study 2 occurred using concurrent timing to implement the two strands during one step, prioritizing them equally and keeping them independent during analysis. The point of integration was during the overall interpretation. In more detail, quantitative data were collected to confirm the results of Phase 1, while additional qualitative data were collected from only MSCCs to develop a more complete understanding of participants’ theoretical perspectives.

The design of Study 3 occurred in one step. Based on the responses of the first two phases, MSCCs were empowered again to offer explanations of the mechanisms behind the opinions of their peers.

In this sequential approach, Study 1 informed Study 2 while Studies 1 and 2 informed Study 3. Below, methods and results are presented per study and in that order.

### 2.1. Materials and Methods: Study 1

#### 2.1.1. Participants

SCCs certified through the CSCCa were recruited via email. Since the project was about the NCAA only, SCCCs/MSCCs whose employers were not members of that association at that time were not included. Based on that criterion, 169, out of 634 SCCs (27%) that agreed to participate, were excluded. SCCs in the sample (*n* = 465) were predominantly white males in their mid-30s (*M*_age_ = 35.85, *SD* = 9.76), possessing a master’s degree, also certified through the NSCA and USA Weightlifting (USAW), working for a Division I institution, and without a military background.

A qualitative component was included to complement and expand upon the results of the quantitative component. Specifically, participants who incorporated MST into athlete exercise regimens were asked to clarify (via written responses to open-ended questions) the specific types of MST they prescribed to athletes. A total of 92 out of the 465 (20%) SCCs indicated they utilized MST. Of that total, 65 (71%) provided a written response to an open-ended question.

#### 2.1.2. Instruments

Data concerning (1) the RQs in the three contexts that SCC programs usually foster MST mentioned above (i.e., visiting military bases, ex-military workshops, own training) and (2) demographic information were collected via a questionnaire (see Appendix A). Before administration, the questionnaire was pilot tested by four SCCs to provide evidence for face and content validity. No issues were reported.

#### 2.1.3. Procedure

After the Institutional Review Board (IRB) approval, the questionnaire was uploaded to Qualtrics (www.qualtrics.com). The participants were then able to go online, sign the consent form, and complete the questionnaire. Recruitment stopped in 8 days and after three reminder emails (every other day).

#### 2.1.4. Statistical Analyses

Quantitative data. Descriptive statistics (e.g., means, standard deviations) were generated to provide summaries of the sample and the responses to the questions. Chi-square tests were performed to determine the significance between MST and sex.

Qualitative data. Thematic analysis, which is an approach for recognizing, examining, and reporting themes within the data [51], was utilized to analyze the SCCs’ responses. This process of analysis includes six steps, as described above by Brown and Clarke (2006). In short, all open-ended responses to the question were organized into an Excel spreadsheet. A qualitative researcher then provided a preliminary review of the data, coded the information, and organized the responses into higher-level themes (categories). From there, the first and third authors reviewed the information as *critical friends* [52]. A critical friend in the current context represents a researcher who provides an alternative perspective and critical evaluation of how a data set has been interpreted. In this sense, the data were first analyzed independently and then jointly. After the data were reviewed and discussed several times, coding was deemed complete when no new themes emerged.

### 2.2. Results: Study 1

#### 2.2.1. Quantitative Data

Firstly, according to SCCs, the majority of their teams do not visit military facilities (97%), invite ex-military on campus for workshops (76%), or prescribe MST in their own training protocols (78%) (see Figure 2). Secondly, it is not their initiative to visit facilities (29%) or invite ex-military (30%), but it is when they are prescribing it (66%) (see Figure 3). Thirdly, in total, (1) one out of five sports that participate is football (see Table 1), (2) around half of those initiatives are taking place during the off-season (see Table 2), and (3) MST is, on average, mostly used in men’s teams ([*Χ*^2^ (1, *n* = 544) = 21.24, *p* < 0.01]). (However, in volleyball only, the results indicated the exact opposite ([X2 (1, N = 39) = 37.38, *p* < 0.01]).) Lastly, MST is used for MT development (military facilities: 60%; workshops: 90%; own training: 88%; Figure 4) and, on average, SCCs believe in the efficacy of those interventions (military facilities: 67%; workshops: 76%; own training: 94%; Figure 5).

#### 2.2.2. Qualitative Data

When asked about what SCCs prescribe when incorporating MST in their own protocols, their responses revealed a distinct emphasis on physical methods. Following are the key elements of the prevailing themes that emerged from the qualitative data. These themes, noted as categories of development, include:Leadership development;Mental development;Physical development;Team development.

Leadership development. The data revealed that participating SCCs incorporated MST to enhance athlete leadership development while they exercise. Leadership is an immensely important component of the success of sport teams and generally involves the process of intentional influence being exerted over others to provide guidance and structure, facilitate relationship formation, and guide coordinated efforts of followers [53,54]. Most participants noted it, either alone or tandem with team development. One participant commented:


*Everything we do is based on military style, because if you are training a team, you better have organization and leadership within your groups.*


Another participant responded:


*Team or small group based conditioning challenges that require someone to take a leadership role.*


Finally, one participant overtly noted how any sort of military training performed with athletes is not done to benefit them physically; instead, the primary aim is to benefit them organizationally.

The specific response was as follows:


*While I generally do NOT aim to incorporate military-style training methods from a physiological standpoint (i.e., long runs carrying a ruck, regular sets of 10–50+ push-ups, standardized sit-up tests, etc.), I do aim to incorporate methods that emulate a military-style from an organizational standpoint. Some specific examples of training practices that we aim to incorporate include warming up in an organized, unified manner, challenging leaders on the team to communicate clearly and with authority, and encouraging athletes to have strong body language (ex: chest up, eyes forward) even when feeling winded.*


Mental development. Next, the data revealed that SCCs incorporated MST to enhance athletes’ mental development while they exercise. Mental development was described with numerous terms and phrases, including “disciplined”, “mental toughness”, “focused”, “decision-making skills”, and “attention to detail”. The cognitive side of MST was clearly important to SCCs. A general sense derived from the data was that the coaches wanted to help their athletes become focused, disciplined, and able to perform when fatigued and under stress.

For example, one SCC responded:


*We do this to build work capacity in a controlled environment and introduce a culture of following instructions and attention to detail.*


Along those lines, a participant commented:


*At the end of training sessions, we do a mental component to see how focused we can be when we are tired. Varies what we will do.*


A different participant provided a specific example of the type of mental training used:

*We regularly (in almost every workout or on-field training session) use a “4-Count Cadence (Counting cadence exercises are a common feature of military training* [55]. *Cadence, or speed, informs the participant how fast or slow an exercise needs to be performed. For example, a four-count cadence exercise would require the leader to count, “One, two, three.”. Then, the participants respond, “One.”. The leader continues, “One, two, three.”. The participants respond, “Two.”. This repeats until the exercise concludes.) Exercise” per US Army standards to reinforce presence of mind and group attention to detail. More of a focus exercise than a physical exercise.*

Additionally, it was not uncommon for participants to reference their military experiences as a rationale for focusing on mental development.

One participant, a veteran, responded:


*I have former experience within US Special Forces working directly with Operational and Performance Psychologists developing metabolic and cognitive programs to increase attention, memory, and decision-making skills with proper conditioning protocols prescribed.*


Physical development. The data also revealed that participating SCCs incorporated MST to enhance athletes’ physical development while they exercise. Though physical development may constitute the most natural connection to MST, it was emphasized about the same number of times as mental development and team development. What is more, the training exercises noted by the participants were often standard fare for collegiate strength and conditioning workouts, including bodyweight circuits, high-intensity interval training, team runs, and bear crawls. Indeed, what exactly constitutes MST appeared unclear to several participants. Many coaches considered their training functional rather than militaristic.

For example, one respondent stated:


*We do things that may be considered military-type training or “functional strength.” Things like tire flips, sled pushing, etc.*


Similarly, a participant noted:


*I’m not totally sure if this meets a definition of “military style training” but for the first 2 weeks on campus our new-comers complete an in-place body weight calisthenics based warm up prior to weight room sessions. Each rep is done on a coach’s cadence and counted out loud by the players. Each movement is performed for 10 reps and must be performed perfectly on cadence to count as a good set. If done unsatisfactorily the round is started over from beginning. The goal is 3 perfect rounds and should take about 8 min when done properly. We do this to build work capacity in a controlled environment and introduce a culture of following instruction and attention to detail.*


A different participant referenced a boot camp for the athletes:


*We have boot camp during preseason with our women’s basketball team. Sometimes we will go out in our sand pit and do some drills like they are in the military. This is probably 4 to 6 total workouts over a 6-week period. Our head coach likes the boot camp stuff this time of year. Normally I wouldn’t go outside and do any of that stuff.*


Team development. Lastly, the data revealed that participating SCCs incorporated MST to enhance athlete team development while they exercise. This was a key theme that was emphasized by numerous coaches. Team development was described in a variety of ways, including “teamwork”, “teams”, “team building”, “communication skills,” “team competitors”, “synchronized movements”, “accountability”, and “one fails we all fail mentality”.

Several participant responses embodied this theme particularly well. For example:


*Team or small group based conditioning challenges that require someone to take a leadership role and rely on the group/team working together to not complete and if possible, win the challenge against the other small group/team competitors.*


Another participant responded:


*Team building exercises that incorporates strategy, teamwork, as well as camaraderie.*


A different participant focused on using training to build teamwork through shared adversity:


*To create an atmosphere of shared misery. Together the team will bond when going through something that is equally tough for everyone.*


### 2.3. Materials and Methods: Study 2

#### 2.3.1. Participants

Only MSCCs were chosen to participate in this phase because their expertise was deemed necessary not only for *member checking* but also for deeper insight. A total of 88 MSCCs agreed to participate in Study 2. Similarly to Study 1, MSCCs who were not employed by an NCAA institution at the time were excluded. As a result, 72 MSCCs (82%) ended up participating.

MSCCs in the sample (*n* = 72) were predominantly white males in their mid-40s (*M*_age_ = 46.13, *SD* = 8.41), possessing a master’s degree, also certified through the NSCA and USAW, working for a Division I institution, and without a military background.

A qualitative component was included so, when merging the two sets of results, we could assess in what ways they converged/diverged from Study 1 and then explain the similarities/differences. Specifically, participants who disagreed with the Study 1 themes (*n* = 8) were given the chance to clarify why. Of that total, five (63%) provided a written response to an open-ended item. Additionally, 66 MSCCs (92%) shared their opinion on the media’s portrayal of the profession and 58 (81%) on ways to prevent future media backlash.

#### 2.3.2. Instruments

Data concerning (1) the emerging RQs in the three contexts mentioned above and (2) demographic information were collected via a second questionnaire (see Appendix A). Before administration, the questionnaire was pilot tested by two SCCs to collect evidence for face and content validity. No issues were reported.

Two items collected quantitative information: one concerned member checking of previous results and the second, their level of agreement with the backlash the profession suffered after the recent incidents described above.

Qualitative data were collected via three items requesting information from:The participants, who did not agree with the themes reported in Study 1, to explain why they did not agree;All the participants concerning what contributed to the media reaction the SCCs suffered;All the participants to identify ways in order to prevent similar criticism in the future.

#### 2.3.3. Procedure

After the IRB approval of the modified proposal (to include this second phase), the questionnaire for Phase 2 was uploaded on Qualtrics. The participants were then able to go online, sign the consent form, and complete the questionnaire. Recruitment stopped after two rounds of data collection separated by 2 weeks (three reminder emails sent out for each round, every other day).

Though helpful, a possible concern about qualitative inquiry is when the researchers both collect and analyze the data. This could give way to research bias [56]. Investigator triangulation, which is the process of employing several different evaluators in a research project, was utilized to enhance the validity of the results and minimize the potential influence of research bias in Study 1.

To further enhance the rigor of the qualitative component of Study 1, a follow-up study was conducted that included member checking and investigator triangulation. Member checking involves the process of actively involving the research participants in the checking and confirmation of the results [57]. In this study, member checking was utilized to explore whether the results and themes generated in Study 1 resonated with SCCs [58]. The methodological value of this approach is objectivism (i.e., validation of previous findings through disconfirming voices) and constructivism (i.e., gaining new insight data via reflection) [59].

#### 2.3.4. Statistical Analyses

Quantitative data. Descriptive statistics (e.g., means, standard deviations) were generated to provide summaries of the sample and the responses to the questions.

Qualitative data. The data collected in Study 2 were analyzed in a manner similar to Study 1. All open-ended responses to the question were organized into an Excel spreadsheet. Next, a qualitative researcher provided an initial review of the data, coded the information, and organized the responses into higher-level themes. Once this was completed, the first and third authors studied the data as critical friends [52]. Coding was deemed finished when no new themes emerged from the researchers’ discussions about the data.

### 2.4. Results: Study 2

#### 2.4.1. Quantitative

The vast majority of MSCCs (*n* = 64; 89%) stated that the themes identified in Study 1 matched their understanding of how SCCs tend to use MST with their athletes. A majority of them (*n* = 39; 54%) thought that the media backlash the profession suffered after recent events in which SCCs were allegedly involved was not justified.

#### 2.4.2. Qualitative

Interpretation of these data collected in response to each of these areas were divided into three content areas:New/additional themes;Contributors to media backlash;Media backlash prevention.

New/additional themes. Minimal insight was gained in terms of which theme or themes might be missing. Of the categories, two participants appeared to question the value of the physical development theme.

Specifically, one MSCC responded:


*Military physical development and playing football are two totally different things. I would not include Physical Development with this style of Leadership Development.*


Contributors to media backlash. A total of 54 MSCCs responded to the question about what they thought contributed to the media backlash. Four themes emerged from the data. Perceived significant contributors to the media backlash about collegiate SCCs utilizing MST were the following:Inexperienced and/or uneducated coaches;Injury and death;Media misinformation and bias;Uninformed outsiders.

Examples of each of these four themes are provided next. The first theme, inexperienced and/or uneducated coaches, was noted by roughly 30% of the participants. It appearred from the data that excessive training and inappropriate application of MST stemming from inexperienced coaches as well as uneducated coaches (who may or may not be experienced strength and conditioning professionals) are believed to be key contributors to the media backlash. The participants also noted that, while they believe most strength coaches are competent, educated professionals, the small number of “bad apples” cause a negative light to be cast on the profession when an athlete gets injured or dies while training.

Specifically, one MSCC responded:


*The hiring of unqualified strength coaches and they hurt the athletes by doing too much too fast and too often.*


Another MSCC noted:


*Poor practitioners within the field that do not demonstrate appropriate professionalism, making it easier to find weakness to attack, especially at the more visible schools.*


Further, sport coaches may be a root cause that leads to the implementation of MST by SCCs. For instance, one participant commented:


*Sport coaches forcing strength and conditioning coaches to incorporate this style of training to help team “mental toughness”. And putting that responsibility on the strength coaches.*


The second theme, injury and death, was noted as an obvious contributor to the media backlash. Best encapsulating this theme is the following response from an MSCC:


*Any time an athlete is seriously injured or dies under a coach’s supervision, that coach/profession is going to be criticized. When there are injuries or death that could have possibly been prevented, the media is going to come after that profession.*


The next theme, media misinformation and bias, highlights a belief among the participants that the backlash stems in part from inaccurate and incomplete reporting as well as media bias, particularly as it pertains to tonality (i.e., bias in how strength coaches and the profession are evaluated and portrayed) and agenda (i.e., bias in terms of how media-preferred issues get dominant media coverage).

In terms of inaccurate and incomplete reporting, one MSCC responded:


*I think that a lack of context and full circumspect understanding of each situation in which military-style training was used is the cause for the media backlash. The media looks for stories and often fabricates and takes information out of context when telling their story.*


Regarding tonality bias by the media, several coaches expressed displeasure in the negative, skewed way in which coaches and the profession are often portrayed by the media.

For example, another participant answered:


*How the media portrays the coaches/staffs in question as uncertified/uneducated and barbaric. Additionally, the media likes to overreact and sensationalize these stories and make it seem like it’s the norm in strength and conditioning.*


As with tonality bias, several coaches blamed agenda bias for the negative backlash. Namely, the media chooses to favor the promotion of negative occurrences within the strength and conditioning profession.

Along these lines, an MSCC observed:


*The media promotes the more animated/aggressive/over the top strength coaches and that’s why they assume the worst.*


Similarly, another participant stated:


*The media is always looking for a negative narrative, placing strength and conditioning professionals in their crosshairs for higher ratings.*


The last theme emerging from the data was uninformed outsiders. Whereas the previous theme about the media focused on how information is presented, this theme centered on individuals outside the strength and conditioning realm found in college athletics. The takeaway from the participants was a sense that many “outsiders” lack a sufficient understanding of strength coaches and the strength and conditioning profession to generate informed opinions when they read or hear sport stories about MST. What is more, a lack of outsider understanding is likely to be compounded in the presence of media misinformation and bias.

Representative of this theme, an MSCC noted:


*Our profession still suffers from stereotypic thinking from those outside the profession, and we have done a poor job offsetting that stereotype.*


Media backlash prevention. In total, 48 MSCCs responded to this open-ended question. Analysis of the data revealed five themes:Strength and conditioning coach education;Media and outsider education;Science-based techniques;Oversight and accountability;Training method avoidance.

Figure 6 provides an illustrative overview of the themes along with representative coach suggestions for preventing backlash in the future.

The first two themes center on education. The participants responded that SCCs as well as the media and outsiders need to be better educated about what is and is not appropriate training for collegiate athletes. Coinciding with education, participants also noted that an emphasis on science-based techniques and greater oversight might mitigate future media backlash.

For instance, one coach responded:


*As a profession we need to be more prudent. Consider all consequences and the perceptions that follow and then weigh the risk to benefits.*


This observation also coincided with the last theme, that of avoiding MST as it is understood by the media and outsiders. The military-style themes of leadership development, mental development, physical development, and team development were generated in Study 1. These themes, which were verified in Study 2, do not represent MST as a form of punishment or a watered-down version of special forces’ training. SCCs indicated they want to develop the whole athlete as well as the team; they want to develop athletes’ minds, bodies, leadership capabilities, and chemistry with teammates. Evident from this final theme is that labeling such efforts as MST, even if only appropriate elements of such training are incorporated into athlete training regimens, may be counterproductive considering how the media and outsiders may interpret such training efforts.

### 2.5. Materials and Methods: Study 3

#### 2.5.1. Participants

Again, only MSCCs were chosen to participate in this phase, for the same reasons as in Phase 2. A total of 127 MSCCs agreed to participate in Study 3. Similarly to Studies 1 and 2, MSCCs, who were not employed by an NCAA institution at the time, were excluded. As a result, 99 MSCCs (82%) ended up participating.

MSCCs in the sample (*n* = 99) were predominantly white males in their late 40s (*M*_age_ = 47.67, *SD* = 6.06), possessing a master’s degree, also certified through the NSCA and USAW, working for a Division I institution, and without a military background.

#### 2.5.2. Instruments

Data concerning the last five RQs were collected via a third questionnaire (see Appendix A). Before administration, the questionnaire was pilot tested by two SCCs to collect evidence for face and content validity. No issues were reported. Therefore, qualitative data were collected via five items.

#### 2.5.3. Procedure

After the IRB approval of the second modified proposal (to include this third phase), the questionnaire for Phase 3 was uploaded on Qualtrics. The rest of the procedure was identical to the qualitative part of Phase 2.

#### 2.5.4. Statistical Analyses

The qualitative data collected in Study 3 were analyzed in a manner similar to Studies 1 and 2. That included organization in an Excel sheet, identification of high-order themes by an author, use of another author as a critical friend, and, finally, coding.

### 2.6. Results: Study 3

This third and final phase included qualitative data only. It was guided by five questions: initiators of MST, reasons for incorporating MST, methods of assessing athletes’ levels of MT, MST in women’s and men’s volleyball teams, and use of psychological interventions. From these questions, various themes were identified and are presented below.

Initiators of military-style training. The responses of the participants (*n* = 60) highlighted that the head sport coaches were responsible for the decision to visit military facilities and/or invite ex-military on campus for workshops.

In support of this, one participant indicated that:


*The coaches of the teams determine this not the strength staff.*


Similarly, another participant stated:


*We don’t but I would imagine a lot of the time it is the sport-coach who initiates a visit.*


Some participants went ahead and offered a possible explanation of that behavior. It was mentioned that head coaches may have a relationship with some military personnel. Therefore, they use this prior relationship to extend an invitation to the military personnel. Additionally, some head coaches may have been exposed to some forms of military training and were eager to involve these as part of their training.

One of the participants noted that:


*In my experiences, generally a position coach has been impressed by or has a relationship with some military personnel and look to bring that element to the training program.*


Athletic directors (ADs) were identified as other initiators. Nevertheless, the number of responses involving ADs in these decisions was much smaller than the ones linking initiation to head coaches.

For instance, one SCC responded:


*Senior Associate Athletic Director for Student-Athlete Wellness/SWA.*


However, on rare occasions, it was noted that the team would suggest the involvement of military personnel in training.

For example, one of the participants explained that:


*Usually the head sport coaches, sometimes the athletes request.*


Reasons for incorporating military-style training on football so often. It should be underlined that, out of the five questions, the participants expanded the most on this item. The coaches (*n* = 39) provided reasons for incorporating MST more into football than other sports. Their responses indicated that football shared similar cultural aspects with the military that are deeply rooted due to exposure from a young age to the idea that football equals war.

One of the participants explained that:


*Because since youth, football coaches have equated football with war.*


Interestingly, that war-like atmosphere is often manifested through the utilization of specific words and phrases. Thus, military language expressions have been adopted and used more often in football than in any other sport.

It was stated by one of the participants that:


*War euphemisms are used often in Football. Going to Battle. Outside the Wire. In the Trenches. Foxhole Guy.*


From the broad theme of shared similarities between football and military training we identified three sub-themes that include:The physical nature of football;Teamwork and communication;discipline and accountability.

Physical nature of football. As with military training, football is a sport that is very physical. In some instances, it has been perceived as a brutal and violent sport. Although football does not deliberately aim to injure or harm players, as a contact sport, its training can be combative, similarly to military training.

To support the above, one of the participants mentioned that:


*The nature of football is that of physicality and violence. I think to some degree it is easy to draw that correlation.*


Another participant noted that:


*You need to be physical and there is a thought that this training will help with that.*


Teamwork and communication emerged as one of the similarities between football and military training. It was believed that the concept of working as a team was an important aspect that is incorporated into football training. Working as a team and effective communication are qualities that are associated with improved performance both in the military and in football.

The participant explained that:


*Football is the closest combative sport there is and it requires a group of young people to have to work together during potential adverse situations.*


Similarly, another participant stated:


*It is a sport that requires a physical development and contact, and a high level of teamwork and communication.*


Coaches also pointed out the similarity in size:


*Positions Groups mimic Military Squad Size.*


Some coaches like the individual/team discipline and character development that come with MST. Football requires discipline and a solid work ethic, such as showing up for practice on time and following the team rules. For a football team to succeed, discipline is essential since it helps athletes focus and work towards achieving their goals. Coaches desire to see in their football teams the same discipline that soldiers have in the military.

It was explained by one of the participants that:


*For the discipline that the military has in their ranks.*


Another coach added:


*Most closely identifies with a large group having one mission and the accountability standards that go with it.*


As part of discipline in the military, one must be accountable to his/her unit and supervisors. This notion includes the importance of starting and completing tasks. In the same vein, football coaches desire players to be present and complete all training exercises.

To support this, it was stated that:


*The experience of the military-style training provides experience of “not giving up” and “finishing” the work out. This attitude has re-appeared in other non-military-style training sessions. The attitude they developed during the military-style training session stayed with them.*


Similarly, another participant stated:


*The military is widely recognized for creating leaders and creating a culture of accountability. Football coaches want to build that into their program, so they look to the military or military style training as means to develop that. I also believe coaches like to see their teams working hard, working together in unison, and being challenged. For better or worse military style training produces this.*


Since the coaches are not using psychological questionnaires, they offered subjective measures of how they know that their MST increases their athletes’ MT levels (*n* = 35). In more detail, coaches are looking at what a mentally tough individual would accomplish to indirectly assess the success of their MST in terms of MT. Some opinions were more general.

One of the participants argued that:


*You cannot objectively know if/how “mental toughness” has been increased. However, through consistent exposure to physical and mental stressors, it can be reasonably assumed tolerance to stressful environments.*


However, some of the participants gave more specific answers. Several examples of manifestations of high levels of MT were offered.

One of the participants explained that:


*If it is successful, I think it shows in the athlete’s resolve or their ability to bounce back after a tough day, week, or rep.*


Similarly, another participant stated:


*We create situations where our student athletes must communicate, think, verbalize cues while being in a fatigued and intense situation.*


Regarding the use of MST more in women’s than in men’s volleyball teams, minimal insight was gained. Out of 34 responses (*n* = 34), most of the respondents did not have a men’s volleyball team and/or an opinion on why that would happen whatsoever. One possible solution was given through a connection with the SCC.

One of the participants explained that:


*Volleyball is usually coached by a higher level strength coach who also works with football.*


In terms of not using psychological interventions, respondents (*n* = 34) defended their behavior by regularly stating that it is out of their scope.

One of the participants explained that:


*We utilize our psychology dept for these interventions.*


Another participant stated:


*We handle the physical. There are other departments that handle the mental aspect.*


Two other sub-themes were evident in their responses. Coaches offered the lack of time and limited knowledge as secondary possible explanations.

One SCC wrote:


*Just because there are really great empirically-based psychological interventions out there, doesn’t mean that I have the time to implement them. I only have 4–6 h per week with the athletes in the off-season.*


Another participant specified:


*We should, but probably aren’t researched enough to know what to do.*


## 3. Discussion

This study aimed to add evidence towards the development of cultural best practices in US collegiate athletics. For the first time, perceptions of a purposive sample of key stakeholders, who claim to specifically prioritize the student-athlete’s overall health and well-being, were utilized.

By recruiting certified SCCs through CSCCa, we conducted a study using a three-phase (*n* = 465; *n* = 72; *n* = 99), mixed-method design and collected data to first answer eight guiding RQs (Study 1) and then, two (Study 2) and five (Study 3) emerging RQs. Below, based on thοse results, we provide general per RQ inferences and then specific per stakeholder inferences.

### 3.1. General Discussions

The results showed undoubtedly that MST is not popular at all in the NCAA. This disproves the belief communicated by the media that MST is widespread in the NCAA.

Unless SCCs are choosing to prescribe it, it is not their initiative. This is not surprising in terms of SCCs choosing to prescribe MST in their own training. What is of more interest is the finding that others, even if it is still relatively unpopular, promote visiting military facilities and/or inviting on campus ex-military for workshops. The head sport coach was identified as the main initiator.

MST is more popular in football. This finding is in accordance with recent research on the higher prevalence of incidents in that sport [60,61]. Their responses indicated that the idea that football equals war is somewhat a tradition in that sport. More specific explanations included similarities in physicality, teamwork/communication, and discipline/accountability.

Except for volleyball, there is a consistent connection between MST and male sports. This does not come as a surprise because football (a male-only NCAA sport) incorporates MST the most. The finding on women’s volleyball supported further investigation; however, the coaches did not provide noteworthy insight. The only other female sport with similar findings was rugby, in which *sanctioned aggression* is valued in terms of motivation [62].

MST is more popular during the off-season. This result is in accordance with the time of the year when the majority of those events take place. It is also in line with the part of periodization, when less sport-specific practices take place, collegiate SCCs spend a lot of time with their players (e.g., countable hours), and they are in charge of the conditioning program [63].

Whether for leadership, mental, physical, and/or team development, the vast majority of SCCs use physical protocols in contrast to psychological ones. Previous research confirms our current findings [47,48,64]. In addition, SCCs are trained to develop more physical than psychological training protocols [65]. Out of scope, limited time, and knowledge comprised their explanations for not choosing psychological protocols. These exact themes were identified in previous research [48].

The vast majority of MSCCs perceive the Study 1 themes as accurate in terms of how SCCs tend to use MST with their student-athletes. This methodological technique of member checking offered objectivity by having these key stakeholders validate the previous findings.

Whenever MST is chosen, it is used for MT development. Our findings are in accord with the literature as collegiate SCCs have been found to use physical training for developing MT [47,48,64].

SCCs believe in the efficacy of those interventions. That efficacy is measured through subjective opinions of “mentally tough behaviors” (e.g., focus, mindfulness, facing adversity/stress/difficulties, bouncing back, completing tasks). These examples appear to have many similarities with the key dimensions of the Mental Toughness Index (MTI) [66]. They also support a recent recommendation that mindfulness should be examined as another dimension of MT [67]. Our results are in agreement with the literature, as collegiate SCCs have been found to be confident in developing efficacious MT through intentional physical practices [47,48,64]. A recent meta-analysis raised concerns as a limited number of studies have used physical training only and the overall quality of observational and pre-post experimental designs was not high [68].

A majority of MSCCs believe that the media backlash towards the profession was not justified. MSCCs think that inexperienced and/or uneducated peers, the severity of certain events, media misinformation/bias, and uninformed outsiders most contributed to the media backlash the profession suffered. As a proactive remedy for the future, MSCCs think collegiate SCCs should be accountable, educate themselves and others (media and outsiders), and develop/update best practices based on science and holistic training and not on training methods that are unfamiliar with or could be considered as out of scope.

### 3.2. Specific Discussions

*Culture* is defined as “shared elements that provide the standards for perceiving, believing, evaluating, communicating, and acting among those who share a language, a historic period and a geographic location” ([69] (p. 408)). It is important to understand that some of these elements are passed down to the next generation of athletes and coaches as “unexamined assumptions”([69] (p. 408)).

This study was not uncritically assuming or attempting to argue that adversity/*trauma* is always detrimental [70], that hypermasculine subcultures have absolutely nothing positive to offer to athletes [71], or that MT is always bad [72,73,74]. However, in our case of possible sport subcultures allegedly putting young collegiate athletes in danger, it is imperative to use the coaches’ perceptions to establish best cultural practices and offer specific suggestions to coaches, media, and athletes.

#### 3.2.1. Coaches

For the first time ever, SCCs have evidence to demonstrate that they do not frequently use MST and that the backlash from the media was perceived as unfair. In addition, MST is more popular in men’s sports and, more particularly, in football. It would be valuable for SCCs to try to recognize the cultural mechanisms behind MST and specific sport/sex, especially when literature just recently identified associations between this type of environment and sex/sport [75]. Furthermore, SCCs believe that their specific MST protocols develop MT. However, recent research [48] confirmed previous findings [47] that SCCs still are unsure about what constitutes MT and hardly assess the effectiveness of their (mainly physical) training methods using psychometrically sound instruments. Moreover, what SCCs operationally described as MST does not appear as something out of context or scope. Lastly, SCCs confirm the presence of a potential ethical blurring [76] and appear mindful of the existence of inexperienced and/or uneducated peers. As a result, they ask for accountability, education [48], and the development of best practices based on a holistic approach [77].

#### 3.2.2. Media

It is evident that MST is not reportedly popular at all in the NCAA. SCCs, without denying potential issues with hypermasculine (football) subcultures that their field needs to address internally, clearly reject the way they are portrayed in and by media (e.g., using unaccustomed exercise and promoting physical exertion beyond exhaustion and fatigue). Although they understand the high level of public interest in a story covering a catastrophic event, they accuse the media of bias and misinformation. Although SCCs confirm that “events” do happen under their supervision, they underline the very low frequency and that the initiative is typically the head coach’s. SCCs operationally defined MST as (mainly physical) protocols that lead towards leadership, mental, physical, and/or team development, which appears to be in accordance with standards of practice. In their opinion, they accomplish a lot more; however, that is not conveyed in those media stories.

The media are an undeniable key player in the sports industry. They are used for information, education, and entertainment purposes, and all three sectors of sport (i.e., public sector, nonprofit sector, and professional sport) need them to survive and prosper in the 21st century [78]. The media should perhaps perform their own constructive criticism and identify what role they would like to play in the development of cultural best practices. On average, the SCCs are clearly not satisfied with their conduct so far; however, they are willing to inform them on their responsibilities and skills. Even if we accept the notion that *identity* is the manifestation of culture [75], there is a clear misalignment between self-perception (i.e., SCCs) and outside perception (i.e., media) of the SCCs’ identity that needs to be addressed.

#### 3.2.3. Athletes

There are more than half a million student-athletes competing in NCAA championships [79]. Therefore, the implications of collegiate sport cultures affect a substantial number of young adults.

Student-athletes should expect (1) to go through a to-a-certain-degree unpleasant process as they try to fit into a new (suboptimal or even optimal) sporting environment [80], (2) to need even higher levels of conformity, if they move to the professional level [81], and (3) MT to lead to different outcomes based on the context (positive psychology constructs are not unilaterally positive/negative) [82]. However, our results show that, on average, current NCAA student-athletes should not expect to find themselves in subcultures that use MST and/or do not approach strength and conditioning training in a holistic way.

According to SCCs, MST is unusual in the NCAA. Our findings do not support the frequency of the association of SCCs and MST that the media have reported or the necessity for calling for more regulation of the profession. Even when incorporating MST in their own strength and conditioning protocols (most likely in football or in a men’s sport during the off-season), the occurrence is low. This evidence indicates that, although media reports may be accurate on the severity of those incidents, the backlash to the profession of strength and conditioning coming from those anecdotally based statements might not be reasonable and/or objective. This is further supported by the fact that SCCs are not the ones who propose visiting military facilities or hosting workshops with ex-military. SCCs, when prescribing MST in their own regimens, do not appear to advise in favor of out-of-scope training (i.e., physical MST protocols aiming for leadership, mental, physical, and/or team development), are aware of areas they should improve, are willing to work on them, and, in general, tend to protect their athletes’ holistic development.

Overall, student-athletes should be aware of cultural best practices and continue to strive for improvements. However, our findings do not indicate that they should be worried about the current practices. Of note, even in terms of cardiovascular etiologies, highly competitive NCAA institutions (e.g., A5DI) implement rigorous screenings [83].

### 3.3. Limitations

Alongside the findings of this study, there are a number of limitations to be acknowledged. Therefore, the authors suggest caution when interpreting the results based on the following:Data were collected via questionnaires. Although evidence for validity was collected, conclusions were drawn based on self-reported data. While a popular method of data collection, it may be biased/misleading [84].Although we asked the NCAA for information concerning the characteristics of their SCCs, we did not receive any. Therefore, we cannot be sure that our sample is representative of the whole strength and conditioning population employed by institutions that are NCAA members. Heterogeneity/representativeness constitutes a possible threat to external validity [85].A few items received a low response rate. Low power, as a result of a small sample size, is identified as a statistical conclusion validity issue (threat causing overly conservative bias) [85].Researcher bias is possible, given that the first author is a CSCCa Written Examination Committee member and an SCC. Researcher bias could lead to several issues, including sampling and interpretation bias [86].

### 3.4. Conclusions and Future Research

For the first time, this research sheds light on a current issue that affects not only the student-athletes’ safety and well-being but also the reputation of the vocation of the strength and conditioning as a whole. Through multiple phases of quantitative and qualitative data collection, integration, and interpretation, our findings add to cultural best practices by describing the current status of strength and conditioning training in the NCAA and by raising awareness on the future role of each key stakeholder. More specifically, it can be concluded that on average: (1) MST is not a popular strength and conditioning method in the NCAA; (2) when MST takes place, it is in the form of physical, in-scope protocols; (3) the current culture promoted by the NCAA SCCs does not indicate transmission of values for any unethical/unhealthy expectations; and (4) coaches, athletes, administrators, and media should continuously strive to stay informed and educated with the intention of safeguarding our youth.

However, additional research will be needed. For instance, in order to generalize these findings, SCCs certified by other professional organizations (e.g., NSCA) should be recruited in the future, and similar efforts should take place in different settings, such as in high schools or the National Association of Intercollegiate Athletics (NAIA). In addition, it should be investigated further why (1) their supervisors (e.g., head coaches, ADs) foster MST and (2) SCCs are usually the only ones mentioned in those negative media reports. Lastly, it is important in our future research efforts on cultural best practices to overall differentiate between *multiculturalism* and *cross-cultural psychology*. While multiculturalism deals more with the interaction among various cultures within one setting, cross-cultural psychology can be framed as “comparisons across cultures or countries, as opposed to comparisons of groups within one society” ([87] (p. 15)). For instance, cultural best practices in collegiate athletics should be investigated as to how the individual cultural differences of athletes of the same team interact in the same cultural microsystem/team (multiculturalism) but also as cultural comparisons between different sports/sexes within one cultural macrosystem/association (cross-cultural psychology). This is also supported by recent MT research in Australian rules footballers that recognizes the implications of the chosen *unit of analysis* (individual vs. group) [88]. Of note, cultural sport psychology (CSP) needs to continue to encourage both researchers and practitioners to prioritize culture [89].

## Figures and Tables

**Figure 1 sports-10-00092-f001:**
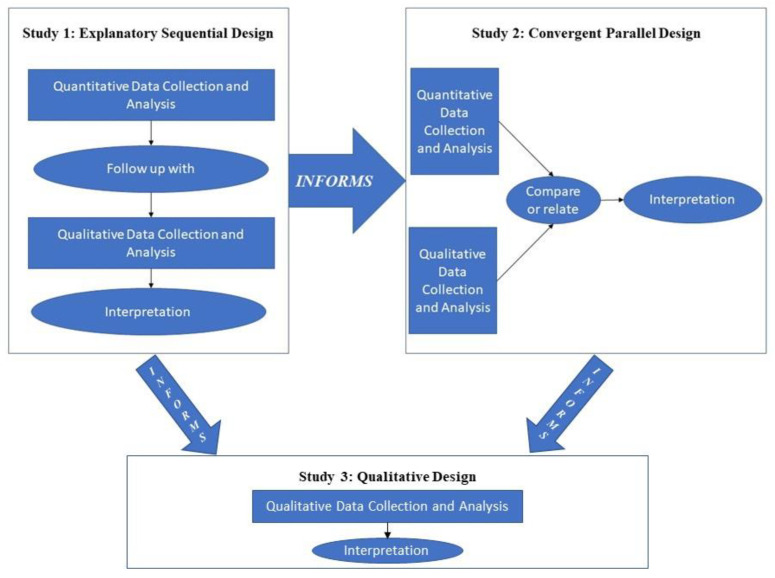
Our overall three-phase flowchart depicted through two sequential mixed-method designs and one qualitative design.

**Figure 2 sports-10-00092-f002:**
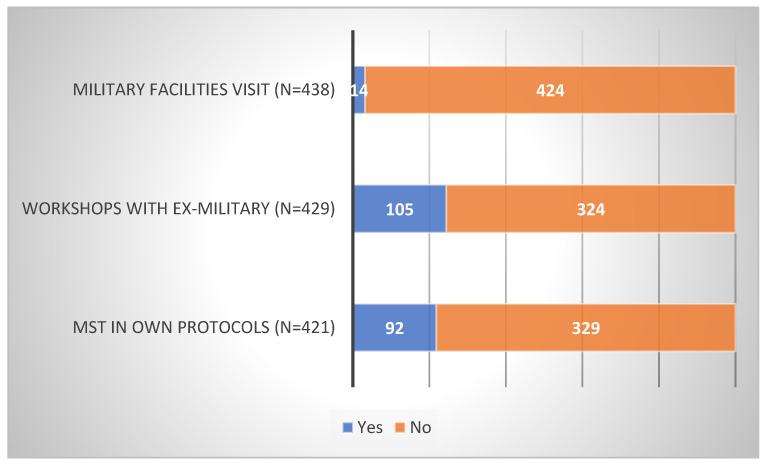
Data on popularity of MST in the NCAA.

**Figure 3 sports-10-00092-f003:**
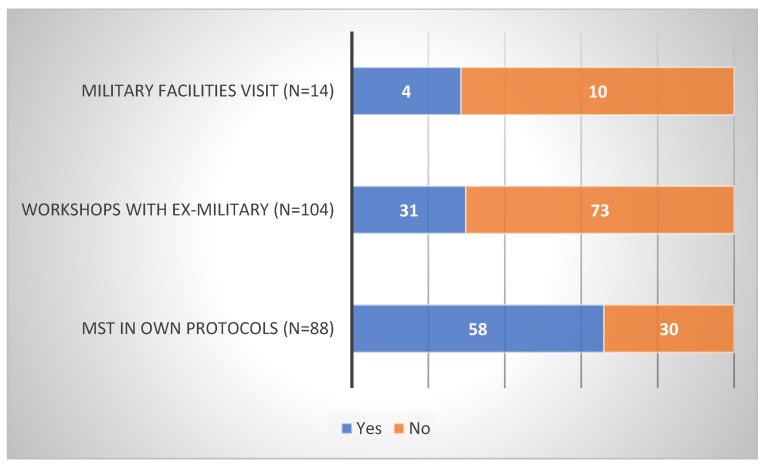
Data about MST as an initiative of SCCs.

**Figure 4 sports-10-00092-f004:**
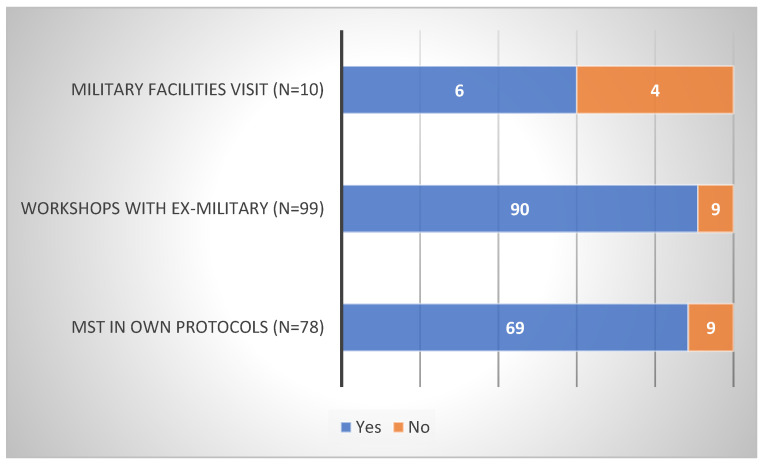
Data on MST being used for MT development.

**Figure 5 sports-10-00092-f005:**
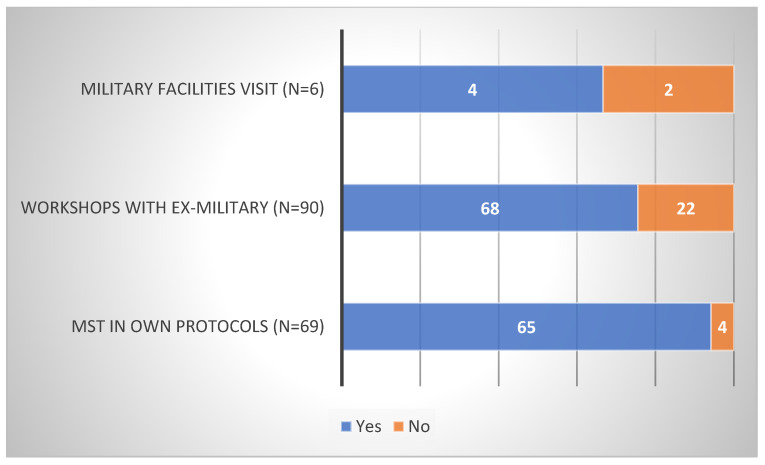
Data on MST being efficient in MT development.

**Figure 6 sports-10-00092-f006:**
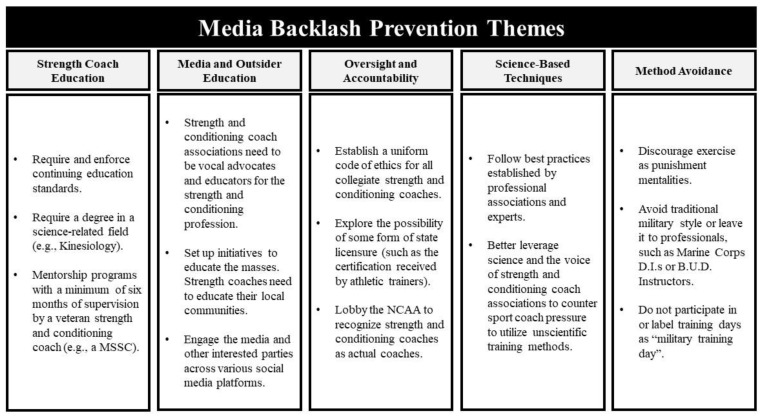
Media backlash prevention themes.

**Table 1 sports-10-00092-t001:** Data on NCAA sport(s) that participate in MST (in alphabetical order).

	Military Facility Visit (*n* = 45)		Workshops with Ex-Military (*n* = 282)		MST in Own Protocols (*n* = 198)	
Sport	Count	Percentage	Count	Percentage	Count	Percentage
Baseball (M)	3	6.67	20	7.09	22	11.11
Basketball (M)	3	6.67	23	8.16	16	8.08
Basketball (W)	1	2.22	22	7.80	19	9.60
Beach Volleyball (W)	0	0.00	2	0.71	0	0.00
Bowling (W)	0	0.00	2	0.71	1	0.51
Cross Country/Track and Field (M)	2	4.44	7	2.48	5	2.53
Cross Country/Track and Field (W)	2	4.44	6	2.13	5	2.53
Fencing (M)	0	0.00	12	4.26	7	3.54
Fencing (W)	0	0.00	1	0.35	0	0.00
Field Hockey (W)	0	0.00	1	0.35	0	0.00
Football (M)	9	20.00	56	19.86	42	21.21
Golf (M)	1	2.22	3	1.06	5	2.53
Golf (W)	2	4.44	3	1.06	5	2.53
Gymnastics (M)	1	2.22	1	0.35	0	0.00
Gymnastics (W)	0	0.00	1	0.35	0	0.00
Ice Hockey (M)	0	0.00	6	2.13	3	1.52
Ice Hockey (W)	0	0.00	3	1.06	1	0.51
Lacrosse (M)	1	2.22	6	2.13	2	1.01
Lacrosse (W)	1	2.22	7	2.48	4	2.02
Rifle (M)	2	4.44	1	0.35	0	0.00
Rifle (W)	2	4.44	1	0.35	0	0.00
Rowing (W)	1	2.22	4	1.42	2	1.01
Skiing (M)	0	0.00	1	0.35	0	0.00
Skiing (W)	0	0.00	1	0.35	0	0.00
Soccer (M)	1	2.22	9	3.19	4	2.02
Soccer (W)	2	4.44	15	5.32	7	3.54
Softball (W)	1	2.22	17	6.03	7	3.54
Swimming and Diving (M)	1	2.22	5	1.77	4	2.02
Swimming and Diving (W)	1	2.22	5	1.77	4	2.02
Tennis (M)	2	4.44	5	1.77	4	2.02
Tennis (W)	1	2.22	5	1.77	5	2.53
Volleyball (M)	0	0.00	5	1.77	1	0.51
Volleyball (W)	2	4.44	15	5.32	16	8.08
Water Polo (M)	1	2.22	1	0.35	0	0.00
Water Polo (W)	0	0.00	3	1.06	0	0.00
Wrestling (M)	2	4.44	7	2.48	7	3.54

Note. M = men; F = women.

**Table 2 sports-10-00092-t002:** Data on time of sports year MST is implemented.

	Military Facility Visit (*n* = 17)		Workshops with Ex-Military (*n* = 145)		MST in Own Protocols (*n* = 134)	
Time of Season	Count	Percentage	Count	Percentage	Count	Percentage
Post-season	2	11.76	9	6.32	18	13.43
Off-season	8	47.06	85	58.62	72	53.73
Pre-season	4	23.53	39	26.90	27	20.15
In-season	3	17.65	12	8.28	17	12.69

## Data Availability

The data presented in this study are available on request from the corresponding author.

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
