# Peer review of "Mental Toughness Development via Military-Style Training in the NCAA: A Three-Phase, Mixed-Method Study of the Perspectives of Strength and Conditioning Coaches"

_sports, 2022, doi:10.3390/sports10060092_

Round 1

Reviewer 1 Report

The manuscript shows an interesting and complex topic. The background needs to be improved looking for claryfing concepts. The methodology is good, but it is neccessary to explain more details about the procedure. The results are interesting.

Title

  • Too long. “Mental Toughness Development via Military-Style Training in the NCAA: A Three-phase, Mixed-method Study of the Perspectives of Strength and Conditioning Coaches”. In my opinion, it is better a short one. Maybe it is not necessary to write the method.

Abstract

  • Please review and summarize the manuscritp. For example:
    • Sport cultures transmit values for anticipated conduct. Have you talk about it in the document?
    • Recent events have resulted in injuries/deaths of National Collegiate Athletic Association (NCAA) student-athletes, usually during off-season football training. Is this enough as a scientific approach to the problema?
    • Strength and conditioning coaches (SCC) have been allegedly involved by incorporating military-style training (MST). Are there scientific evidences?
    • Where is the objective?
    • Have you recap the results from the questionaries?

Introduction

  • Lines 39-41. The approach to the problem needs to be based on scientific reference. “Although the cause could be multi-factorial and unique to each case (e.g., primordial factors, appropriate medical coverage, acclimatization), SCCs have been allegedly implicated [9-11]”. Both references are from press, not from journals. And, if this is the real problem, it is necessary to search academic references.
  •  
  • Line 93. Purpose of the Study. If the main purpose was “to contribute to cultural best practices in sports” (lines 94-95), why it is not included in the title and in the introduction? What are the best cultural practices? If you focus your attention in these practices, you can analyze which ones are not good… such as MST and probably more. Then, you can write about the speficic objective: “exploration of the use of MST in the NCAA (e.g., sport, sex, period of year, specific activities), its connection to SCCs, its association with MT development, and the role of media”. To recap: clarify the objective and/or the introduction (sequencing ideas).

Method

  • Methodology is good in using a mixed method and selecting to the participants.
  • Line 100. About the tool: Do the guiding Research Questions (RQs) a scientific background? How has it been created? Has it been validated by an experts? in phases 2 and 3, who and how have the new questions been created?
  • Line 170. About the participants, is the sample significance?
  • In the qualitative analysis, how have the codes been créate? Have you shown them? Can you explain better the procedure: how many researchers have participated in the analysis, how have you measure the agreement between them?

Quality of Presentation: the manuscript has been written in an appropriate way.The estructure and the guidelines are appropiate.

Scientific Soundness: the study is interesting and the method is ok but it is necessary to explain with more details the tool (how is has been created) and the analysis (specially qualitative ones).

Interest to the Readers: the paper can be read by coaches, athletes, researchers and people interested in this topic.

In my opinion, it can be submitted to the Journal with some changes.

Reviewer 2 Report

General comments

Through this study the authors wanted to contribute to cultural best practices in sports and, specifically, to investigate the confirmation and exploration of the use of military-style training in the National Collegiate Athletic Association, its connection to strength and conditioning coaches, its association with mental toughness development, and the role of media. The study was performed in three phases with a mixed-method design.

The study presents a considerable amount of work. Despite the limitations, well described by the authors, I think this is an important and different study for this field of research.

Specific comments

Abstract

The abstract clearly summarizes the entire manuscript.

Introduction

The introduction summarizes the literature clearly following the logical thread: Known → Unknown → Research Question. The gap in the literature that the authors would like to fill has been described.

Materials and Methods

The (quali-quantitative) methodology is clearly explained.

Results

Results clearly show descriptive statistics and questionnaire responses (qualitative data).

To fix:

-Line 211, replace with “Results: Study 1

Discussion

The discussions clearly explain what was found. The results have been interpreted correctly. The limitations are clearly described. The conclusions are justified. The take-home message is clear.

To fix:

-Line 671: Replace with “General Discussions”.

-Line 726: Replace with “Specific Discussions”.

-Line 820: Replace with “Conclusions and Future Research”. In the first part,  restate your main findings and the novelty of the paper according to the current literature to help better readers understand how this paper is different from others already published. Then continue with Future Research.

Round 2

Reviewer 1 Report

Thank you for your explanations, some of them has clarified my suggestions